# Thrombin-Mediated Formation of Globular Adiponectin Promotes an Increase in Adipose Tissue Mass

**DOI:** 10.3390/biom13010030

**Published:** 2022-12-23

**Authors:** Peter Zahradka, Carla G. Taylor, Leslee Tworek, Raissa Perrault, Sofia M’Seffar, Megha Murali, Tara Loader, Jeffrey T. Wigle

**Affiliations:** 1Department of Physiology and Pathophysiology, University of Manitoba, Winnipeg, MB R3E 0J9, Canada; 2Department of Food and Human Nutritional Sciences, University of Manitoba, Winnipeg, MB R3T 2N2, Canada; 3Canadian Centre for Agri-food Research in Health and Medicine, St. Boniface Hospital Albrechtsen Research Centre, Winnipeg, MB R2H 2A6, Canada; 4Department of Biochemistry and Medical Genetics, University of Manitoba, Winnipeg, MB R3E 0J9, Canada; 5Institute of Cardiovascular Sciences, St. Boniface Hospital Albrechtsen Research Centre, Winnipeg, MB R2H 2A6, Canada

**Keywords:** adiponectin, globular adiponectin, thrombin, cleavage-resistant mutant, obesity, adipocytes

## Abstract

A decrease in the circulating levels of adiponectin in obesity increases the risk of metabolic complications, but the role of globular adiponectin, a truncated form produced by proteolytic cleavage, has not been defined. The objective of this investigation was to determine how globular adiponectin is generated and to determine whether this process impacts obesity. The cleavage of recombinant full-length adiponectin into globular adiponectin by plasma in vitro was used to identify Gly-93 as the N-terminal residue after proteolytic processing. The amino acid sequence of the cleavage site suggested thrombin was the protease responsible for cleavage, and inhibitors confirmed its likely involvement. The proteolytic site was modified, and this thrombin-resistant mutant protein was infused for 4 weeks into obese adiponectin-knockout mice that had been on a high-fat diet for 8 weeks. The mutation of the cleavage site ensured that globular adiponectin was not generated, and thus did not confound the actions of the full-length adiponectin. Mice infused with the mutant adiponectin accumulated less fat and had smaller adipocytes compared to mice treated with globular adiponectin, and concurrently had elevated fasting glucose. The data demonstrate that generation of globular adiponectin through the action of thrombin increases both adipose tissue mass and adipocyte size, but it has no effect on fasting glucose levels in the context of obesity.

## 1. Introduction

Adiponectin, also known as adipocyte complement-related protein (Acrp), is a 30 kDa protein secreted by adipocytes that promotes metabolic homeostasis and exerts a protective effect on the vasculature [1]. Adiponectin also functions as an insulin sensitizing protein and exhibits anti-inflammatory properties [2]. Reduced production of adiponectin is thus associated with greater insulin resistance and an increased risk of diabetes, both of which increase the incidence of cardiovascular disease [3,4]. 

Serum adiponectin levels are lower in obese individuals than in persons of normal weight [5], as well as those with metabolic syndrome and Type 2 diabetes mellitus [6,7]. A causal relationship between hypoadiponectinemia and insulin resistance was established via generation of adiponectin knockout mice [8,9,10]. Moreover, Maeda et al. [9] observed that a high-fat diet increased the severity of insulin resistance. Regardless of the underlying cause, decreased adiponectin signalling appears to be important in the onset of insulin resistance, especially in the context of obesity [4,11]

Fruebis et al. [12] were the first to report the presence in serum of low levels of a truncated 18 kDa form of adiponectin that they termed globular adiponectin (gAcrp) since it contains the complement factor C1q-like region. Both Fruebis et al. [12] and Waki et al. [13] subsequently showed that proteolytic cleavage of adiponectin in vitro with either trypsin or leukocyte elastase generates a proteolytic fragment that physically and pharmacologically resembles gAcrp. Various studies have compared the biological actions of fAcrp and gAcrp in vitro [14,15,16,17]; however, animal experiments have been incapable of defining the specific functions of the full-length adiponectin protein (fAcrp) and gAcrp in vivo due to the confounding presence of gAcrp in all experiments involving fAcrp. Thus, while previous investigations have reported some cases that the actions of gAcrp are identical to those of fAcrp [18], in other circumstances, the effects were the opposite of those seen with fAcrp [19,20]. Consequently, a definitive direct comparison of these two proteins has not been achieved in vivo. 

To address this knowledge gap, we proceeded to develop an fAcrp analogue that could not be cleaved to gAcrp. As part of this process, conclusive identification of the site at which the endogenous protease cleaves gAcrp was required. The cleavage site, which was recognized by thrombin, was subsequently mutated and the recombinant mutant protein infused into adiponectin-knockout mice made obese by high-fat feeding for comparison with mice infused with recombinant gAcrp. The data indicate that gAcrp specifically increases adipose tissue mass and adipocyte size but does not affect fasting glucose levels.

## 2. Materials and Methods

Preparation of Human Recombinant Proteins: fAcrp was cloned, expressed and purified from bacterial lysates by Ni^2+^-agarose chromatography as previously described [21]. An R92G mutation of adiponectin was prepared by site-directed mutagenesis. The primers used for this purpose were: forward 5′GGAATGACAGGAGCGGAAGGGCCAGGGGGCTTTCCCGG-3′; reverse 5′-CCGGGAAAGCCCCCTGGCCCTTCCGCTCCTGTCATTCC-3′. The mutation was confirmed by sequencing (University of Calgary). The mutant protein, designated mAcrp, was prepared as previously described for the wild-type protein [21]. The gAcrp sequence (starting at G93 in fAcrp) was amplified using the forward primer 5′-GGCTTTCCCGGAACCCCTGGCAGGAAAGG-3′ and reverse primer of 5′GCGGATCCTCAGTTGGTATCATGGTAGAG-3′, sequenced and subcloned into the pET45b expression vector. The protein was expressed and purified as described for fAcrp and mAcrp, except that extraction of the protein from inclusion bodies [22] was required before Ni^2+^-agarose chromatography could be performed.

fAcrp Cleavage Assay: Adiponectin (1 µg) was added to a reaction mixture (total volume 10 µL) consisting of 8 µL α-MEM (minimal essential medium) and 0.5 µL rat plasma, a commercial product prepared by pooling samples from control animals (a gift of Dr. Paul Fernyhough, University of Manitoba). The assay was conducted for 30 min at 37 °C. After addition of 2× sample buffer (125 mM Tris-HCl pH 6.8, 2% sodium dodecyl sulfate, 20% glycerol, 2% β-mercaptoethanol) and heating at 95 °C for 5 min, the samples were analyzed by SDS-polyacrylamide gel electrophoresis (15% acrylamide). Detection of the cleavage product was achieved either by Western blotting (as described in [23] using a human-specific primary antibody) or by direct protein staining with Gel Code Blue (Thermo Fisher Scientific, Mississauga, ON, Canada). In the latter case, the gel was placed into Gel Code Blue for 60 min and then destained in water overnight. Quantification of band intensities was carried out using a GS800 Imaging Densitometer (Bio-Rad Laboratories, Mississauga, ON, Canada) and the data are expressed as arbitrary units [23].

Gel filtration: A 0.2 mL aliquot of commercial rat plasma was applied to an Ultrogel AcA 44 (Sigma Aldrich Canada, Oakville, ON, Canada) column (4.2 mL bed volume; 130 kDa exclusion limit) and eluted with 10 mM Tris-HCl pH 8, 50 mM NaCl, 0.1 mM EDTA. Fractions of 160 µL were collected once elution was started. The cleavage activity in 1 µL of each fraction was measured with the standard assay as described above.

gAcrp Sequencing: Adiponectin was incubated with plasma under the standard assay conditions and the cleavage products were separated by SDS-polyacrylamide gel electrophoresis on a 15% Tris-HCl Ready Gel (Bio-Rad Laboratories). The running buffer (125 mM Tris, 0.959 M glycine, 0.5% SDS) was prepared with UltraPure water (Cayman Chemical, Ann Arbor, MI, USA). For mass spectrometry sequencing, the gel was rinsed with UltraPure water and sent to The Manitoba Centre for Proteomics and Systems Biology (University of Manitoba). For Edman sequencing, the proteins were transferred to Sequencing Grade polyvinylidene difluoride (PVDF) membrane (Bio-Rad Laboratories) using transfer buffer consisting of 10 mM CAPS pH 11, 10% methanol and UltraPure water. The membrane was subsequently washed for 5 min with UltraPure water, stained for 5 min with 20 mL Gel Code Blue and then destained with 50% analytical grade methanol prepared in UltraPure water. The membrane was sent to Advanced Protein Technology Centre Peptide Sequencing Facility, The Hospital for Sick Children (Toronto, ON, Canada).

Cell Culture and Western Blot Analysis: Smooth muscle cells from were prepared from explants of porcine coronary artery according to standard procedures [24]. Cells were maintained in serum-free supplemented media for 4 days before treatment. Recombinant proteins were added directly to the cell medium. Cells were lysed with 2× sample buffer and Western blotting was carried out as previously described [23]. Primary antibodies (phospho-p38 MAPK, β-tubulin) were from Cell Signaling (New England Biolabs Ltd., Pickering, ON, Canada). Quantification of band intensities was carried out using a GS800 Imaging Densitometer (Bio-Rad Laboratories) and the data are expressed as arbitrary units relative to loading control as previously described [23].

Infusion of recombinant adiponectin: Adiponectin knockout mice (APN-KO [10]) were generously provided by Philipp Scherer (University of Texas Southwestern Medical Center, Dallas, TX, USA) and a colony established in the R.O. Burrell Animal Facility of the St. Boniface Albrechtsen Research Centre. Forty male APN-KO mice were placed on a high-fat diet (60% kcal from fat; D12492, Research Diets, New Brunswick, NJ, Canada) when they reached 8 weeks of age and maintained on the diet for 8 weeks. At that time, the mice were randomly placed into 1 of 4 groups (*n* = 10 mice/group): (i) control, (ii) fAcrp, (iii) gAcrp, (iv) mAcrp. All groups were implanted with a model 1004 (0.11 µL/h, 28-day capacity) Alzet micro-osmotic pump (DURECT Corporation, Cupertino, CA, USA) that was connected via a catheter to the jugular vein to ensure direct delivery into the bloodstream. The pumps for the control mice contained saline, whereas the other groups received pumps with the respective recombinant protein prepared in saline. The pumps delivered 5 µg of protein per day over the 28-day infusion period as reported previously [21]. All mice were kept on the high-fat diet for the entire infusion period. Mice were weighed before and 28 days after implantation of the pumps, with correction made for pump weight. The protocol was approved by the University of Manitoba Animal Care Committee and followed the guidelines of the Canadian Council of Animal Care.

Tissue Collection and Analysis: Twenty-six days after infusion was started, the mice were fasted for 6 h and tail vein blood glucose was measured using an Alpha TRAK blood glucose monitoring system (calibrated for rodents) and Alpha TRAK2 blood glucose test strips (Abbott Laboratories, Chicago, IL, USA). On day 28 of the infusion period, the animals were euthanized and epididymal, peri-renal and inguinal adipose tissues were collected and weighed. A portion of epididymal adipose tissue was frozen in CryoGel (Electron Microscopy Sciences, Hatfield, PA, UK) in a dry ice-ethanol bath. Sections of 8 µm were prepared with a cryotome and digital images captured for adipocyte size measurement as previously described [25]. 

Statistics: All in vitro conditions were tested in triplicate (*n* = 3), whereas analyses of the in vivo study utilized *n* = 10 per group, except for adipocyte size, which was based on *n* = 5 per group. Data meeting the assumptions of normality and homogeneity were analyzed by ANOVA followed by Duncan’s multiple range test for post hoc testing. Data that were not normal or homogeneous were analyzed by non-parametric testing with the Kruskal–Wallis test and significant differences were determined by least significant difference test. The level of significance was set at *p* ≤ 0.05. Analyses were performed using the Statistical Analysis System (9.4M7, SAS Institute, Cary, NC, USA) software program. 

## 3. Results

*fAcrp is cleaved by incubation with plasma*: It has been reported that the formation of gAcrp is mediated by trypsin and leukocyte elastase in vitro [12,13]; however, the identity of the enzyme responsible for generating globular adiponectin in vivo is not known. Since gAcrp is present in the circulation [12], it is plausible that a protease located in plasma catalyzes this reaction. To test this hypothesis, recombinant fAcrp and plasma prepared from rat blood were incubated together for 30 min at 37 °C in cell culture medium, with the expectation that this would most closely emulate the conditions found in the circulation. Analysis of the samples by SDS polyacrylamide gel electrophoresis revealed that a novel band of approximately 18 kDa appeared after incubation with plasma. This band was detected by Western blotting (Figure 1A) with an antibody selective for both human fAcrp and gAcrp [14], thus indicating it is derived from adiponectin. Due to the specificity of the antibody for human adiponectin, endogenous adiponectin in the rat serum was not detected. Moreover, this band could also be seen upon staining for total protein (Figure 1B). While staining for total protein revealed numerous bands originating from bacterial and plasma sources that were unrelated to adiponectin, the presence of recombinant fAcrp (input) and gAcrp (product) was unambiguous. For this reason, we utilized total protein staining for the majority of our experiments.

*Assessment of fAcrp cleavage by trypsin and elastase:* It has been documented that both trypsin and leukocyte elastase can cleave adiponectin, resulting in the formation of a product that is physically and functionally similar to globular adiponectin. For comparison, we tested both of these proteolytic enzymes in the standard assay and observed formation of a band of approximately 18 kDa as expected (Figure 1C,D). The band pattern, however, was not identical to that obtained with plasma (Figure 1A,B).

*Serum cleavage activity has unique properties*: We subsequently investigated the properties of the plasma cleavage activity to obtain data that could help with its identification. Maximum adiponectin cleavage was obtained with 0.5 µL of plasma, and the reaction was completed within 15 min (Figure 2A,B). The presence of saturable activity suggested the cleavage factor was likely a protein. Interestingly, cleavage was dependent upon the presence of divalent cations, particularly calcium, since the activity was blocked by addition of the chelators EDTA (broad specificity) and EGTA (calcium selective) (Figure 2C). We therefore assessed whether additional calcium or magnesium beyond the amounts present in the α-MEM component of the assay (final concentration in assay: calcium = 1.2 mM, magnesium = 0.8 mM) would enhance the activity. No increase in proteolysis was observed with either magnesium or calcium, although magnesium at concentrations above 2 mM inhibited the reaction (Figure 2D).

To determine the size of the protein responsible for adiponectin cleavage, plasma proteins were separated by gel filtration chromatography and the fractions monitored with the standard assay. It was observed that formation of globular adiponectin occurred only with fractions 27 to 33 (Figure 2E). These results indicate the size of the native protein is greater than 80 kDa, as indicated by the elution profile of creatine phosphokinase. Based on these data as well as the results of the size exclusion chromatography which indicated the protease was significantly larger than either trypsin (23 kDa) or leukocyte elastase (32–36 kDa), it was concluded another protease was responsible for formation of globular adiponectin in vitro.

*Identification of fAcrp cleavage site*: To determine the enzyme class of the protease, globular adiponectin was generated by incubation with plasma and the reaction product was sent for sequencing using both mass spectrometry and Edman degradation approaches. Mass spectrometry resulted in the sequencing of several unique peptides, all of which could be located within fAcrp. The amino terminal sequence of the peptide that was nearest to the N-terminus of fAcrp was Gly-Phe-Pro-Gly-Thr-Pro-Gly-Met and began with Gly-93. Since the data were insufficient to establish that this peptide came from the N-terminus of the 18 kDa band, a second sample was sent for Edman sequencing. The first amino acid detected with this method was ambiguous, with Ser, Gly, Ala and Met as the possible amino acids for residue 1; however, the amino acids for residues 2 to 11 were unambiguous: Phe-Pro-Gly-Thr-Pro-Gly-Arg-Lys-Gly-Glu. This matched with the results of the mass spectrometry sequencing, thus confirming that the N-terminal amino acid of the 18 kDa cleavage product is Gly-93. This position does not match with the cleavage sites for either trypsin or leukocyte elastase, as indicated in Figure 3.

*Identification of the serum protease*: We conducted a survey for possible proteinases capable of cleaving the plasma-sensitive sequence identified in adiponectin using the MEROPS database [26]. Searching with the sequence Pro-Arg-Gly-Phe (P2-P1-↓-P1′-P2′ format) returned 5 hits: trypsin, kallikrein-related peptidases 5, 6, 14 and thrombin. The sequence specificity for these enzymes is Lys/Arg (P1) for trypsin, Arg (P1) for the three kallikrein-related peptidases and Pro (P2) Arg (P1) Gly/Ala/Ser (P1′) for thrombin. Given the broad specificity for both trypsin and kallikrein-related peptidases, plus the fact circulating levels for these enzymes are low under normal conditions, we elected to focus on thrombin as a likely candidate for the adiponectin cleavage factor.

To determine whether thrombin is responsible for the cleavage of adiponectin by plasma, we tested the effect of three inhibitors, SSR69071, hirudin and argatroban. SSR69071 is a selective inhibitor of leukocyte elastase. Although the sequence specificity of leukocyte elastase (elastase-2) does not conform to the adiponectin cleavage site (P1 = Tyr/Ala), the effect of SSR69071 was tested because leukocyte elastase has been proposed as the most likely candidate to cleave adiponectin in vivo [13]. However, when SSR69071 was added to the cleavage assay, formation of globular adiponectin was not blocked (Figure 4A). In contrast, both thrombin inhibitors, hirudin and argatroban, effectively inhibited the production of the 18 kDa fragment (Figure 4B,C). These results strongly suggest that thrombin is the protease responsible for the catalytic cleavage of adiponectin into globular adiponectin.

*Thrombin cleaves fAcrp* in vitro: To examine directly whether thrombin is capable of generating globular adiponectin, thrombin was used in the cleavage assay instead of plasma. A 30 min incubation with thrombin was sufficient to completely convert the input 30 kDa protein into two fragments of 18 kDa and 12 kDa (Figure 4D). The 18 kDa band migrated at the same position as the band that was obtained with plasma, but the 12 kDa band did not appear in the reactions using plasma. Since plasma contains a wide variety of proteins, including other proteases, it is likely that this fragment is rapidly degraded and therefore is not detected. Cleavage by thrombin is concentration dependent (Figure 4E) and is inhibited by both hirudin and argatroban (Figure 4F). These results support the conclusion that thrombin is responsible for the formation of globular adiponectin in vitro. 

*Effect of cleavage site mutation on gAcrp formation:* To confirm the sequence specificity of fAcrp cleavage by thrombin, we altered the cleavage site sequence from arginine at position 92 to glycine. The mutated protein (R92G = mAcrp) was prepared and tested in the cleavage assay with both plasma and thrombin. Under optimal conditions for cleavage of the wild-type protein by thrombin, neither plasma nor thrombin produced the 18 kDa gAcrp product, although trypsin was able to cleave the mutant protein (Figure 5A). Since it is possible the mutation altered the structure of the protein and in this way interfered with the actions of thrombin, we compared the ability of the wild-type and mutated proteins to stimulate smooth muscle cells, which respond differently to fAcrp and gAcrp [14]. Addition of either protein to porcine smooth muscle cells activated p38 MAPK with similar potency while gAcrp had only minimal effect (Figure 5C,D). These results indicate the mutation does not affect the structure of the protein enough to prevent it from binding to its receptor and firmly establish that cleavage of fAcrp by thrombin occurs between Arg-92 and Gly-93.

*Distinguishing the physiological effects of fAcrp and gAcrp* in vivo: The availability of mAcrp makes it possible for the first time to directly assess the physiological actions of the full-length form of adiponectin in vivo without the confounding presence of gAcrp. Specifically, infusion with mAcrp is equivalent to treatment with the only the full-length protein, since mAcrp cannot be cleaved into gAcrp. In contrast, infusion with fAcrp, which can be cleaved, is equivalent to treatment with a mixture of fAcrp and gAcrp. APN-KO mice were chosen as the model to eliminate the effects of endogenous adiponectin (fAcrp + gAcrp). To determine whether adiponectin would correct a metabolic dysregulation it was necessary to do the experiment in APN-KO mice challenged with a high-fat diet, since these mice otherwise show no phenotype [9,10].

Recombinant fAcrp, gAcrp and mAcrp were individually infused by osmotic mini-pump into APN-KO mice that had been previously on high-fat diet for 8 weeks to induce obesity. There was no difference in weight gain during the 4 weeks after administration of the recombinant proteins relative to vehicle (saline) control (Figure 6A). However, compared to the vehicle control and mAcrp groups, fAcrp and gAcrp administration increased the epididymal, peri-renal and inguinal fat pads relative to body weight (Figure 6B–D), thus leading to an increase in total body fat (Figure 6E). This change in fat mass is most likely driven by enlargement of adipocyte size in response to fAcrp and gAcrp (Figure 6F), which would be expected if there was an increase in lipid storage. Overall, these results suggest that gAcrp is responsible for adipose tissue expansion due to accumulation of triglycerides in the enlarged adipocytes. In contrast, as demonstrated by the mAcrp group, full length adiponectin does not affect adipose tissue properties. It is likely that gAcrp is highly potent since the small amount formed during infusion of fAcrp was sufficient to produce the same effect as gAcrp alone. 

The changes in fat pad size are indicative of an alteration in lipid metabolism induced by gAcrp. Interestingly, measurement of fasting glucose levels 2 days before study termination showed that gAcrp infusion did not affect this parameter relative to the control group (Figure 7). Rather, fasting glucose levels were significantly elevated in response to both fAcrp and mAcrp. While mechanisms cannot be advanced based on these data, this infusion study clearly establishes that fAcrp and gAcrp have distinct physiological effects in vivo on adipose tissue handling of lipids and glucose metabolism under obesogenic conditions.

## 4. Discussion

This is the first report to unequivocally establish that gAcrp has distinct biological activities in comparison to fAcrp in vivo. This was achieved by identification of both the protease responsible for producing gAcrp endogenously from fAcrp, namely thrombin, and the site at which thrombin cleaves fAcrp. With this novel information, we designed a mutant form of fAcrp that cannot be converted into gAcrp yet retains the ability to stimulate p38 MAPK activation. This unique tool, mAcrp, was subsequently used to compare the physiological actions of fAcrp and gAcrp on adipose tissue in APN-KO mice in the context of obesity. Upon infusing the recombinant proteins individually into obese APN-KO mice, fAcrp and gAcrp increased adiposity and adipocyte cell size but mAcrp did not. In contrast, mAcrp and fAcrp increased fasting glucose levels, while gAcrp did not. These results indicate that fAcrp has distinct biological effects from gAcrp with respect to both lipid and glucose metabolism under conditions where gAcrp is not present (i.e., when non-cleavable mAcrp is used in place of thrombin-sensitive fAcrp); thus, it may be concluded that the generation of gAcrp from fAcrp explains the similar responses to fAcrp and gAcrp infusion that were observed in vivo. 

An adiponectin cleavage product (gAcrp) was first reported by Fruebis et al. [12], and since that time, only trypsin and leukocyte elastase have been shown to generate an adiponectin fragment in vitro that resembles gAcrp in both size and function [12,13]. However, whether these proteases or another enzyme generates gAcrp in vivo has never been conclusively established. To address this gap, we developed a novel assay to screen for the presence of a protease capable of cleaving fAcrp, since formation of gAcrp is presumed to occur in the circulation. Our findings indicate that thrombin is the only activity present in plasma capable of generating gAcrp. The requirement for calcium, sensitivity to thrombin-specific inhibitors and apparent large size by gel filtration, which can vary from ~80 kDa in its native form (with subunits of 36 kDa) to over 200 kDa when present in large protein complexes [27], all support the conclusion that production of gAcrp in plasma is mediated by thrombin, and not by trypsin or leukocyte elastase. Furthermore, we also identified the N-terminus of plasma-generated gAcrp and showed it is in a unique location that is not recognized by either trypsin or leukocyte elastase or another protease (Figure 3). 

While attempts to compare the physiological actions of fAcrp with gAcrp have been made, these have for the most part been performed in mice expressing endogenous fAcrp [12,28,29,30,31,32] rather than in adiponectin knockout mice [33,34]. Experiments not involving adiponectin knockout mice are further confounded by the fact fAcrp is converted to gAcrp by an endogenously circulating enzyme. By preparing mAcrp, a unique tool that cannot be converted to gAcrp (Figure 5A) and retains the ability to signal as demonstrated by activation of p38 MAPK (Figure 5C,D), it became possible for us to examine for the first time the individual contributions of fAcrp and gAcrp in vivo.

The gAcrp employed in the current study was designed to initiate at Gly-93, the site of thrombin cleavage (Figure 3), and is distinct from the recombinant gAcrp used in most other studies which initiate at Gly-105 of the human sequence or Met-111 of the mouse sequence. These latter two positions in the human and mouse sequences are located between the trypsin and leukocyte elastase cleavage sites mapped by Fruebis et al. [12] and Waki et al. [13], respectively (Figure 3). In contrast, we have identified Gly-93 as the N-terminus of the gAcrp generated by plasma cleavage of fAcrp (Figure 3). It is worth noting that the sequence flanking the cleavage site (Gly-87 to Gly-96) is 100% conserved across species (including human, mouse, dog, horse, opossum, goose, chicken), which suggests there is a critical biological function associated with fAcrp cleavage across evolution.

Our data indicate that fAcrp and gAcrp have differing biological effects on adipose tissue in the context of obesity. Specifically, our results demonstrate that gAcrp and fAcrp, but not mAcrp, increases both lipid storage in adipose tissue and adipocyte size without affecting total body weight. This finding demonstrates that for adiponectin to alter lipid storage in vivo it is necessary for fAcrp to be converted into gAcrp since the full length, uncleavable form of adiponectin (mAcrp) did not alter lipid storage. In contrast, animal studies in which gAcrp has been either expressed genetically or infused into the circulation have shown that this form of adiponectin has beneficial effects on body weight and adipose tissue dysfunction [12,29]. The different conclusions reached by other investigators, however, were based in most cases on results obtained with model systems that were not free of the confounding presence of both forms of adiponectin. Furthermore, as we have shown in Figure 6, differences were detected between fAcrp and mAcrp that are most likely due to the presence of gAcrp, which would be generated in the fAcrp-treated but not mAcrp-treated animals. These results therefore suggest that the presence of gAcrp is sufficient to alter the biological outcome obtained with fAcrp, which infers there might be differences in the relative potencies of fAcrp and gAcrp, and furthermore, that both the N-terminal portion of the protein and the globular domain have distinct biological functions. 

Our results indicate that gAcrp promotes both fat pad size and adipocyte hypertrophy. It is likely that both processes are closely linked, however, at this time it cannot be determined which serves as the primary target. However, an increase in adipocyte size is a strong indication that gAcrp stimulates greater lipid accumulation, whether by de novo lipid synthesis or enhanced uptake from the circulation. Furthermore, gAcrp-induced adipocyte hypertrophy is likely an outcome of its ability to inhibit adipogenesis [35]. Although gAcrp has been shown to indirectly influence overall energy homeostasis [12], a direct effect is more likely to explain the increase in fat content based on the occurrence of hypoadiponectinemia and metabolic dysfunction in a human population carrying a genetic mutation in the globular domain [36]. At the same time, the lower efficacy of mAcrp in promoting adipocyte hypertrophy, which was 50% that of gAcrp but also not statistically different from the control, may indicate fAcrp can also affect fat mass, but it is likely through induction of adipogenesis [37].

In comparison with adipose tissue mass, gAcrp has no effect on fasting glucose levels. This contrasts with the elevation in fasting glucose induced by treatment with both fAcrp and mAcrp. These divergent results indicate that fAcrp and gAcrp have opposing effects on these facets of metabolism. Our findings support the model that the ability to increase fasting glucose is lost when fAcrp is converted into gAcrp. This definitive statement is made possible through our utilization of cleavage-resistant mAcrp. This form of adiponectin, which cannot be converted into gAcrp, has the same effect as fAcrp but not gAcrp. This unique tool will be valuable in future studies that expand on the findings reported herein, especially those that examine other metabolic aspects not assessed in this study. For example, is the increase in fat mass balanced by a loss in muscle mass given that body weight does not change? Similarly, are the differences in fasting glucose level attributable to changes in whole body metabolism or to alterations in the metabolism of specific tissues? More directed studies are needed to address these questions.

At the same time, our identification of thrombin as the likely candidate for cleavage of fAcrp in vivo implies that gAcrp may only be formed at times that thrombin itself is activated, which would be in response to endothelial damage or in prothrombotic states such as obesity and diabetes [38,39]. In the context of obesity, thrombin activation has been shown to cause weight gain [40], and suppresses fAcrp expression through a tissue factor-mediated process [41]. Our results imply that gAcrp formation may have a role in these events, possibly by mediating the actions of thrombin on adipocytes. However, the mechanism by which this occurs, particularly as it applies to lipid accumulation by adipose tissue, remains unclear. Future progress in this direction will require an assay specific for gAcrp that will allow the determination of circulating gAcrp levels in various disease conditions in relation to their thrombotic and inflammatory status.

The close relationship between adiponectin and obesity has provided a foundation whereby adiponectin mimetics are being developed for the purpose of treating weight gain and the resultant metabolic complications [42,43,44]. The major limitation in the development of these mimetics is the lack of clarity regarding the most appropriate form of adiponectin to employ since this is a determining factor for selecting the target receptor. Our findings suggest that fAcrp does not impact adipose tissue expansion but may influence glucose levels. On the other hand, gAcrp increases fat pad size and adipocyte size, and the latter indicates the development of adipocyte dysfunction. Therefore, it may be more appropriate to have a therapeutic which antagonizes AdipoR1 for which gAcrp is the primary high-affinity ligand [28].

Determination of the endogenous protease responsible for generating gAcrp enabled us to design a study capable of addressing for the first time the issue of physiological relevance as it applies to adiponectin cleavage. The novel findings reported open new avenues for the study of gAcrp, its relevance in health and disease, and its possible role in prothrombotic settings such as obesity and other metabolic diseases which typically increase the risk of cardiovascular disease.

## Figures and Tables

**Figure 1 biomolecules-13-00030-f001:**
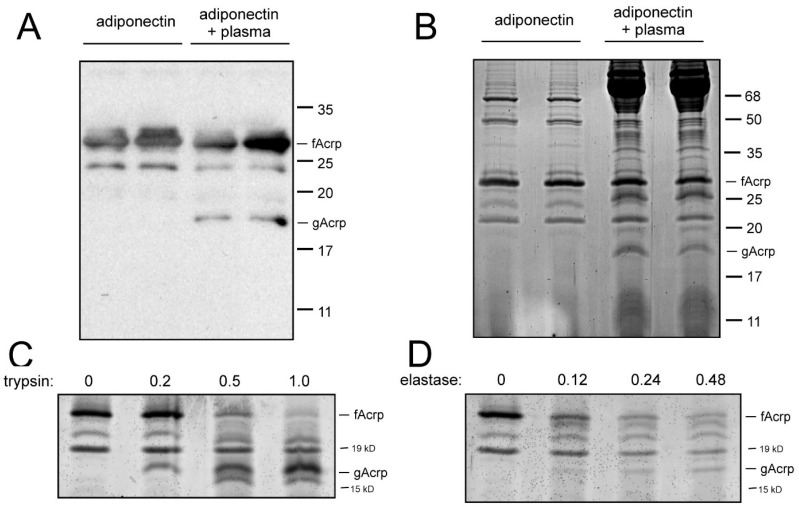
Cleavage of adiponectin by plasma in vitro. Recombinant adiponectin (1 µg) was incubated in duplicate in the absence or presence of 0.5 µL plasma for 30 min as described in Methods. (**A**) Adiponectin and the cleavage product were detected by Western blotting. (**B**) Total protein in the samples was detected by staining the SDS/polyacrylamide gel with Gel Code Blue. In both panels, the positions of full-length (fAcrp) and globular (gAcrp) adiponectin are indicated, as are the positions of molecular mass markers run in parallel. (**C**,**D**) Sensitivity of adiponectin to proteolytic cleavage. Recombinant adiponectin (0.4 µg) was incubated as described in Methods, with varying amounts (in µg/mL) of either trypsin or leukocyte elastase. Adiponectin cleavage was monitored by staining with Gel Code Blue. The positions of full-length (fAcrp) and globular (gAcrp) adiponectin are indicated, as are the positions of molecular mass markers run in parallel.

**Figure 2 biomolecules-13-00030-f002:**
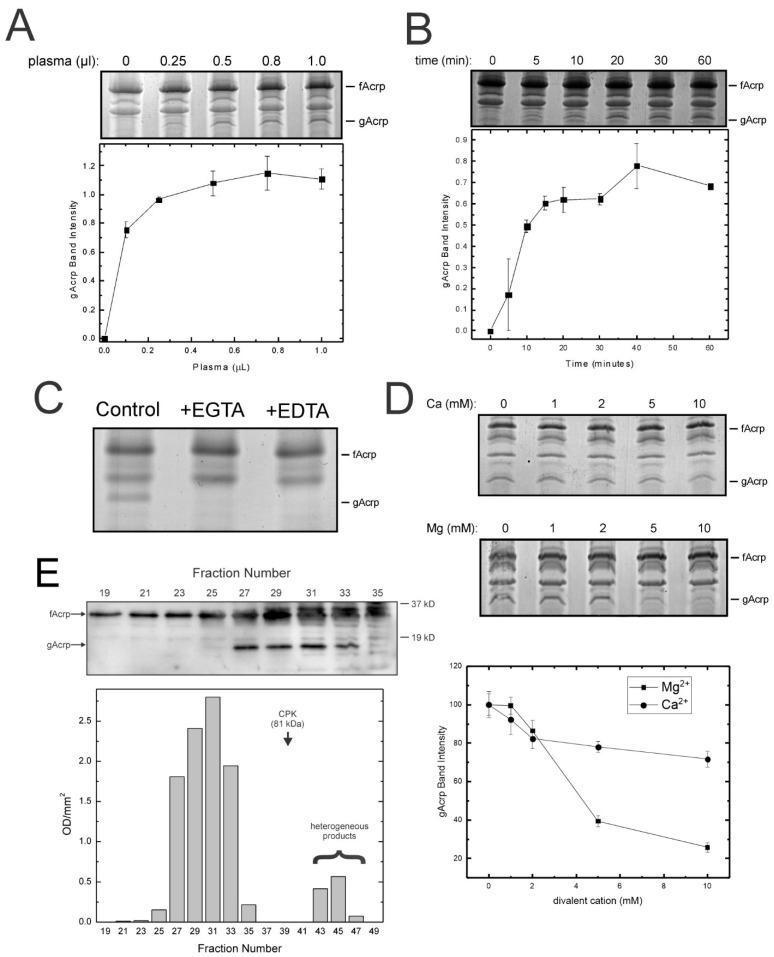
Catalytic and physical properties of the plasma gAcrp-generating activity. Cleavage of 1 µg adiponectin by plasma was examined in relation to amounts of plasma (**A**), incubation time (**B**) and presence of chelators of divalent cations (**C**), as well as varying concentrations (**D**) of calcium and magnesium. Conditions were based on the standard assay described in Methods. (**A**–**D**) representative SDS-polyacrylamide gels stained with Gel Code Blue. (**A**,**B**,**D**) the intensity of the 18 kDa band was quantified by scanning densitometry and the results are plotted as means ± sem (*n* = 3). (**E**) Physical properties of the plasma cleavage activity were investigated by gel filtration chromatography with Ultrogel AcA44 as described in Methods. The presence of cleavage activity in the collected fractions was examined with the standard assay. The appearance of a band at 18 kDa was considered indicative of gAcrp formation. Relative activity is plotted as band intensity (OD/mm^2^) obtained by scanning densitometry of the 18 kDa gAcrp band.

**Figure 3 biomolecules-13-00030-f003:**
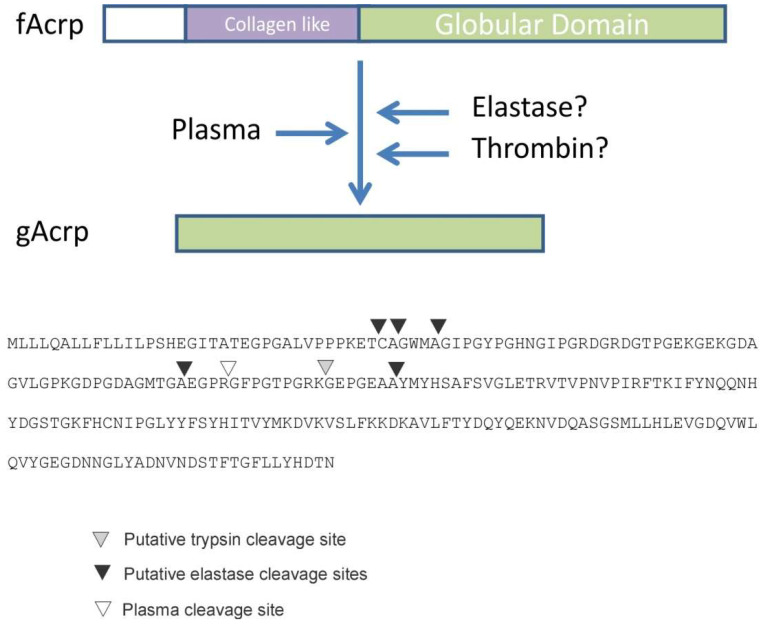
The relative positions of the trypsin, elastase and plasma cleavage sites are indicated on the amino acid sequence of mouse adiponectin. Trypsin and elastase sites were identified from the literature [12,13], while the site cleaved by incubation with plasma was determined with Mass spectrometry and Edman degradation sequencing procedures as described in Methods.

**Figure 4 biomolecules-13-00030-f004:**
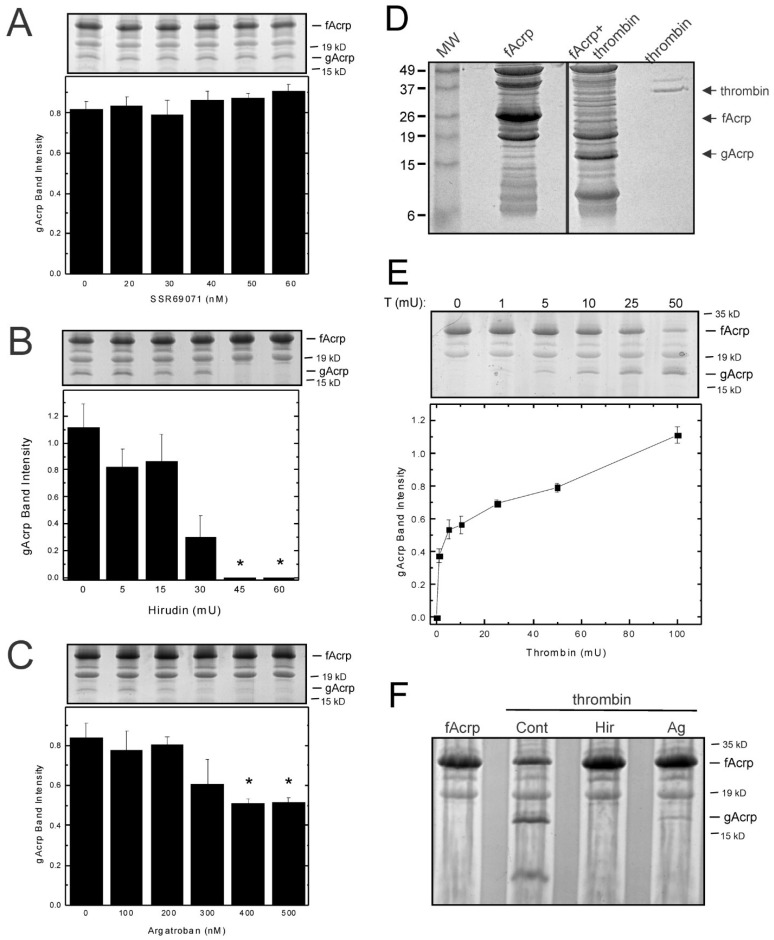
Effect of protease inhibition on cleavage of adiponectin. Cleavage of adiponectin (1 µg) was measured under the standard assay conditions in the presence of varying amounts of (**A**) SSR69071 (leukocyte elastase inhibitor), (**B**) hirudin (thrombin inhibitor) and (**C**) argatroban (thrombin inhibitor). Representative stained gels are provided for each inhibitor. Intensity of the 18 kDa band was quantified by scanning densitometry and the data are plotted as means ± sem (*n* = 3). * significantly different (*p* < 0.05) versus no inhibitor control by ANOVA and Duncan’s multiple range test for post hoc testing. (**D**) To determine sensitivity of adiponectin to cleavage by thrombin, the standard adiponectin cleavage assay was conducted with 50 mU thrombin added instead of plasma. The samples were subsequently examined by staining with Gel Code Blue after SDS-polyacrylamide gel electrophoresis. Molecular mass markers and thrombin were run in parallel as controls. (**E**) Adiponectin cleavage was measured in the presence of varying amounts of thrombin (T). A representative stained gel is shown. Intensity of the 18 kDa band was quantified by scanning densitometry and the data are plotted as means ± sem (*n* = 3). (**F**) Adiponectin cleavage was performed under the standard assay conditions in the presence of 50 mU thrombin in the absence and presence of the thrombin inhibitors hirudin (Hir, 60 mU) and argatroban (Ag, 400 nM). Formation of the 18 kDa product was monitored by staining with Gel Code Blue after SDS-polyacrylamide gel electrophoresis.

**Figure 5 biomolecules-13-00030-f005:**
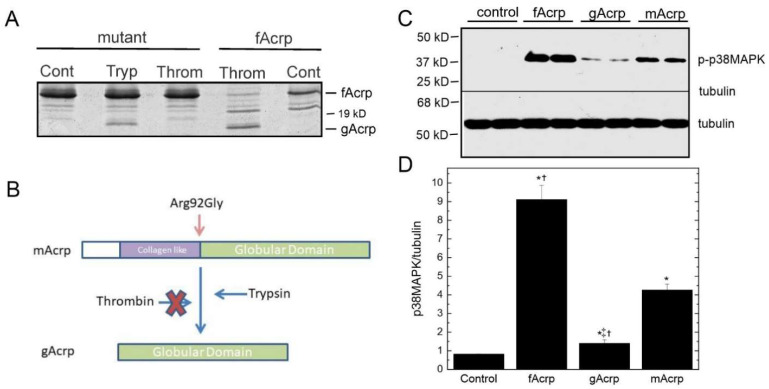
Effect of site-directed mutagenesis that converted Arg-92 to Gly-92 in adiponectin. (**A**) Cleavage of 10 µg mutant adiponectin with 50 mU thrombin and 0.2 µg trypsin was tested under the conditions described for Figure 4. fAcrp served as the positive control. A representative gel is shown. (**B**) Schematic summary of the results shown in panel A. The R92G mutation makes mAcrp resistant to cleavage by serum and thrombin, but not to trypsin. (**C**) Quiescent smooth muscle cells were treated for 60 min with 7 µg/mL fAcrp, gAcrp and mAcrp and subsequently lysed for Western blot analysis of p38 MAPK phosphorylation. A representative blot is shown. (**D**) Band intensities were quantified by scanning densitometry and the data from 3 replicates are plotted (relative to the β-tubulin loading control) as means ± sem. * *p* < 0.05 relative to untreated control; † *p* < 0.05 relative to mAcrp; ‡ *p* < 0.05 relative to fAcrp, by ANOVA and Duncan’s multiple range test for post hoc testing.

**Figure 6 biomolecules-13-00030-f006:**
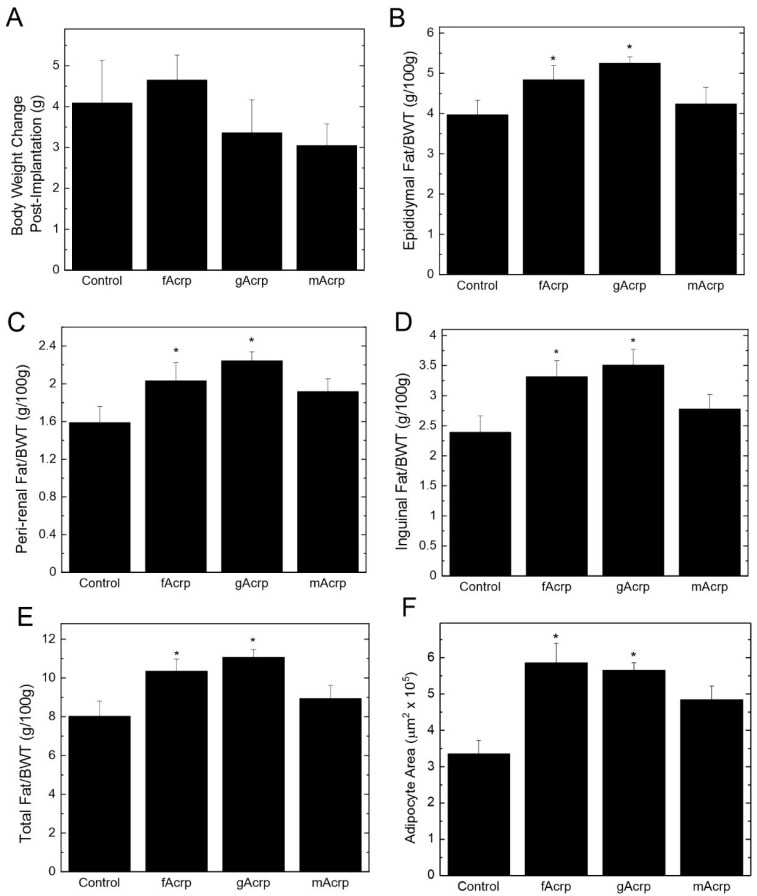
Comparison of fAcrp and gAcrp on body weight and adipose tissue parameters. (**A**) Change in body weight over the 4-week infusion period. (**B**–**D**) Weight of epididymal (**B**), peri-renal (**C**) and inguinal (**D**) fat pads relative to body weight (BWT) at the end of the infusion period. (**E**) Sum of the weights of the 3 major fat pads shown in panels (**B**–**D**) relative to BWT. For panels (**A**–**D**), mean ± sem was determined from *n* = 10 animals per group. (**F**) Adipocyte size was determined using ImageJ; a minimum of 100 cells were analyzed from three images taken of sections of epididymal fat pads. Mean ± sem was based on 5 animals per group. * *p* < 0.05 relative to Control for all panels. Panels (**A**–**E**) by ANOVA and Duncan’s multiple range test; Panel (**F**) by Kruskal–Wallis test and least significant difference test.

**Figure 7 biomolecules-13-00030-f007:**
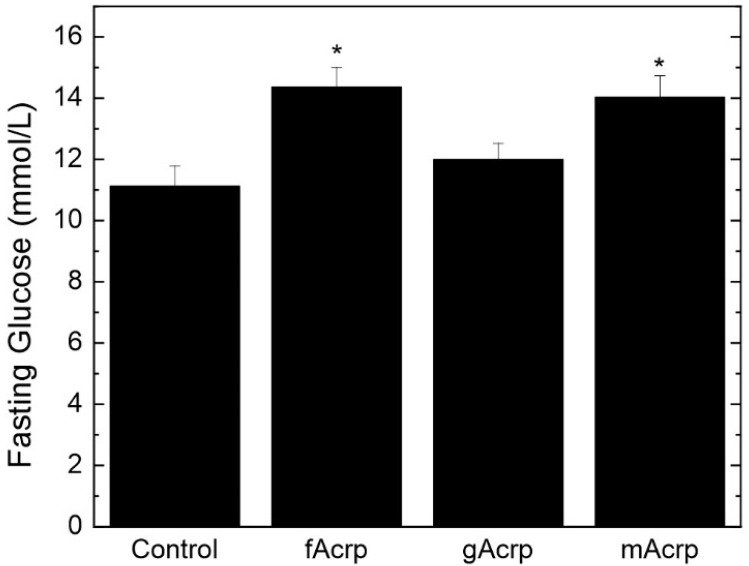
Comparison of fAcrp and gAcrp on fasting glucose levels. Tail vein blood glucose concentrations were measured with an Alpha TRAK blood glucose monitoring system and glucose strips. The values are presented as mean ± sem from *n* = 10 animals per group. * *p* < 0.05 relative to Control as determined by ANOVA and Duncan’s multiple range test.

## Data Availability

The study data are available upon reasonable request.

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
