# Peer review of "Thrombin-Mediated Formation of Globular Adiponectin Promotes an Increase in Adipose Tissue Mass"

_biomolecules, 2022, doi:10.3390/biom13010030_

Round 1

Reviewer 1 Report

The study of Peter Zahradka et al. states that gAcrp has distinct biological activities in comparison to fAcrp in vivo. The authors first identified thrombin as the protease responsible for producing gAcrp endogenously from fAcrp, and identified the site at which thrombin cleaves fAcrp. So they designed a mutant form of fAcrp that doesn't  generate gAcrp.  Finally, using this mutant, gAcrp and fAcrp, they demonstrated that gAcrp increased adiposity and adipocyte cell size in APN-KO mice.

The study was conducted rigorously and in my opinion, requires only a minor revision.

In particular:

Line 51: fAcrp. Since this is the first time that the authors have mentioned fAcrp, I suggest specifying that f means full length

Line 152-153 and fig1B and 4D. Although the presence of gAcrp and fAcrp is unambiguous, the recombinant protein preparation appears to contain several other proteins. The authors should comment this point and produce a SDS page with negative control in order to prove that the additional bands are from E.coli.   Several bands are also present in figure 4D.

Line 293. Globular adiponectin receptors are present on membrane of several cell types, why were smooth muscle cells used? Is there a specific reason?

Line 388-389. “The different conclusions reached by other investigators, however, were based in most cases on results obtained with model systems that were not free of confounding elements”

The authors should better specify what they intend for “confounding elements”. Fruebis' study is cited by many articles, so a more in-depth discussion on this point is important. The authors could also cite studies that confirm their findings albeit in different contexts. (eg Yuchang Fu et al 2005).

Author Response

  1. Line 51: fAcrp. Since this is the first time that the authors have mentioned fAcrp, I suggest specifying that f means full length

Thank you for pointing this out. We have corrected this (line 60) and also made sure gAcrp (line 54) and Acrp (line 40) were also defined

  1. Line 152-153 and fig1B and 4D. Although the presence of gAcrp and fAcrp is unambiguous, the recombinant protein preparation appears to contain several other proteins. The authors should comment this point and produce a SDS page with negative control in order to prove that the additional bands are from E.coli.   Several bands are also present in figure 4D.

Thank you for this suggestion. We had not considered showing a gel stained after extraction of untransformed E. coli just to show the protein pattern from this source. As described in the text (lines 181-183), we felt a comparison of the adiponectin lanes in Figure 1A and 1B is sufficient to show which bands are bacterial in origin. Due to the short time requested for revising the manuscript, we will be unable to complete this request of the reviewer.

  1. Line 293. Globularadiponectin receptors are present on membrane of several cell types, why were smooth muscle cells used? Is there a specific reason?

We have added an explanation for the selection of this cell type to the sentence (line 323).  Specifically, in an earlier published study [14] we showed that smooth muscle cells respond differently to fAcrp and gAcrp, thus making them an ideal model to test mAcrp for activity.

  1. Line 388-389. “The different conclusions reached by other investigators, however, were based in most cases on results obtained with model systems that were not free of confounding elements”

The authors should better specify what they intend for “confounding elements”. Fruebis' study is cited by many articles, so a more in-depth discussion on this point is important. The authors could also cite studies that confirm their findings albeit in different contexts. (eg Yuchang Fu et al 2005).

We thank the reviewer for noticing this. The vagueness of the sentence (line 486-487) has been improved by indicating the source of the “confounding elements.” We have also incorporated the reference of Fu et al (2005) in the Discussion (line 506).

Reviewer 2 Report

1. In Fig. 2E, which fraction number was identical with the molecular size of thrombin?

2. Please add cleavage of adiponectin by thrombin in time dependency.

3. How was the size of adiponectin in the epididymal fat pads after treatment with full-length or globular adiponectin?  Did you look at histological analysis of the epididymal fat by H-E staining?  

Author Response

  1. In Figure 2E, which Fraction number was identical with the molecular size of thrombin?

This is a difficult question to answer. The data available suggest native thrombin is about 80-90 kDa in size (composed of 36 kDa subunits), but it can form aggregates with proteins in plasma as well as other constituents such as cholesterol sulfate, leading to apparent sizes of over 200 kDa. Thus, the native protein would elute with a peak around fraction 37. However, since we fractionated plasma, the higher molecular mass indicated by the peak at fraction 30 is reasonable. This has been made clearer in the Discussion (lines 446-447).

  1. Please add cleavage of adiponectin by thrombin- time dependency to Figure 4.

This would be a nice addition to the figure, however, we are unable to comply within the time frame of this revision. It would take several experiments to determine which thrombin concentration would be the best to use in preparing the final figure.

  1. How was the size of adipocytes in epididymal fat pads after treatment with fAcrp and gAcrp? Did you look at histological analysis of epididymal fat pads by H-E staining?

Information about adipocyte size was provided in Figure 6D of the original manuscript. In that panel, the average area of the epididymal adipocytes was presented. As indicated in the Methods section, the cell size was determined from frozen sections according to a method we published previously [ref 25].  is an excellent question. These data are now shown in Figure 6F of the revised manuscript. Both fAcrp and gAcrp increased adipocyte size 1.75-fold. In contrast, the adipocyte size with mAcrp infusion was not significantly different from control. We feel these results indicate that both forms of adiponectin stimulate adipocyte size in vivo after proteolytic cleavage.

Reviewer 3 Report

I read with great and sincere interest the manuscript devoted to the analysis of the cleavage of 30 kDa Acrp to the globular form. There is no doubt that knowledge of adiponectin processing, but also of the different biological activities of the different forms of adiponectin, is very important with regard to its pleiotropic metabolic effects.

In a series of in vitro experiments, the authors have provided evidence that the globular form of adiponectin is produced by proteolytic cleavage by thrombin, at the GLY93 site. They subsequently prepared a mutant protein resistant to the effects of thrombin and monitored its effects in vivo after infusion into obese adiponectin KO mice. This experiment thus served to distinguish the effects of full length adiponectin from those of globular adiponectin under in vivo conditions. A similar analysis has not been possible before, as full length adiponectin, which has been used in previous studies, could be endogenously cleaved to the globular form.

The article is undoubtedly interesting and, 20 years after the first indications concerning the proteolytic cleavage of adiponectin and the discovery of the circulation of globular adiponectin in a living organism, it comes with new facts that refine and significantly advance the original analysis.

However, the presentation and interpretation of the paper's data leads to a number of issues that should be resolved before the paper is accepted, which I describe below.

1. The authors should document the presence of recombinant forms of adiponectin in plasma of infused mice, as it is important to demonstrate that the mutant adiponectin has not actually been cleaved to the globular form, as cleavage by proteases present in, for example, interstitial fluid, cannot be ruled out. This form of modification is also to be expected in view of the many reports on the biological activity of HMW adiponectin, which is associated with local cleavage of the full length form to the globular form. Both denaturing and native gel, in using an adiponectin specific antibody to detect the protein, should be used to detect the different forms of adiponectin. This would also show whether the recombinant full length and mutant adiponectin can form the usual HMW, MMW and trimeric forms.

2. Analysis of the in vivo effect of recombinant adiponectin provides in fact only very limited insight into the metabolic functions of globular vs. full length adiponectin. The weight of the animals and the two different adipose tissue depots (e.g., the depot representing subcutaneous fat is not selected) does not necessarily mean that the two types of adiponectin have different metabolic functions. The infusion should be followed by at least a determination of a basic metabolic profile of the animals, e.g. at least blood glucose and insulin levels, but preferably GGT.

Since weight did not change significantly, while the weight of the fat depots was altered, this means that body composition must have changed, and it is not discussed. The change in weight must necessarily be accompanied by an analysis of food consumption and should be documented by weight curves for each group, which would also show the same initial weight of the animals in each group.  For Figure 6B and C, which describes the change in depot size, the correction of g to 100 g is given, which makes no sense.

Thus, overall, the experimental part of the in vivo experiments is very vague and its outputs are overstated, and either should be significantly supplemented or preferably not shown. Based on the data presented, it cannot be argued that the “data indicate that gACRP specifically increases lipid accumulation by adipose tissue while fQAcrp does not (line 64-65) or that  Our results indicate that gACRP promotes obesity (line 397)”

3. The authors expressed recombinant proteins in bacteria that are unable to provide the usual posttranslational modifications of adiponectin that affect assembly of the usual forms of adiponectin (Banga, A., et al. (2009). "Adiponectin translation is increased by the PPARgamma agonists pioglitazone and omega-3 fatty acids." Am J Physiol Endocrinol Metab 296(3): E480-489.) Thus, the results should still be verified on proteins expressed in eukaryotic cells

4. Why was rat plasma used when adiponectin processing was tested in vivo in mouse and in vitro in porcine cells? Is it known what thrombin activity the plasma used had (was it a pooled sample, or the plasma was obtained from different animals, in different pathophysiological conditions, e.g. endothelial damage, where a higher thrombin activity is expected)?

5. The text should clearly indicate whether the recombinant adiponectin was human. This information is now hidden in the references to previous work of the group.

6. If the authors performed a comparison of proteolytic activity between plasma, elastase, and trypsin, this result should be demonstrated on a single gel so that it can be compared side by side (Figure 1 A-D). Fig1 C,D-the authors claim that adiponectin was detected by western blotting, but the band pattern is more reminiscent of total protein detection by Gel Code Blue staining. Fig 1 A,B should include a line with plasma alone, where endogenous adiponectin is also present ( albeit at a significantly lower concentration)

7. The identification of thrombin as the major protease of 30 kDa adiponectin was preceded by gel filtration of plasma, which indicated that the protease of interest was greater than 80 kDa. Therefore, trypsin and leukocyte elastase, as well as other possible proteases, were excluded because of their smaller size. However, the size of thrombin also reaches only 37 kDA, as the authors themselves state when presenting the results in Fig. 4D.  Somewhat purposely, they then claim that thrombin is usually in plasma in complex with other proteins- but this may be true for other proteases as well. Moreover, thrombin is most often in complex with antithrombin, which suppresses its proteolytic activity. Although I am not an expert on the coagulation pathway and properties of thrombin, I believe this fact should be mentioned, or it should be acknowledged that the results do not exclude other putative proteases to be able to cleave adiponectin.

8 Figs. 2 and 4 -representative images of the gels do not include some conditions, which are then shown in the graphs showing band intensities. This should be unified. Fig. 2E should be reduced in size to put the entire panel of Fig. 2 on one page.

Author Response

  1. The authors should document the presence of recombinant forms of adiponectin in plasma of infused mice, as it is important to demonstrate that the mutant adiponectin has not actually been cleaved to the globular form, as cleavage by proteases present in, for example, interstitial fluid, cannot be ruled out. This form of modification is also to be expected in view of the many reports on the biological activity of HMW adiponectin, which is associated with local cleavage of the full length form to the globular form. Both denaturing and native gel, in using an adiponectin specific antibody to detect the protein, should be used to detect the different forms of adiponectin. This would also show whether the recombinant full length and mutant adiponectin can form the usual HMW, MMW and trimeric forms.

The reviewer makes an excellent point. Unfortunately, the only blood taken in quantity was at the termination of the study, which was 28 days after implanting the pumps. We have made several attempts to detect adiponectin in these samples, but have so far failed. We suspect the reason is that the reservoir may have emptied by the endpoint and protein turnover reduced the levels below the level of detection. As a result, it is not possible to add this information to the manuscript. Please note, in some respects this was a pilot study, since we did not know what to expect for results. We thus could not entertain all of the possibilities. Instead, we performed key assessments that would guide us in designing the next study. Weekly blood draws to monitor protein levels is part of our future plans. Likewise, we will be able to examine the formation of adiponectin aggregates, although we know fAcrp can form HMW adiponectin as shown in our publication detailing its synthesis [Fuerst et al (2012); ref 14].

  1. Analysis of the in vivo effect of recombinant adiponectin provides in fact only very limited insight into the metabolic functions of globular vs. full length adiponectin. The weight of the animals and the two different adipose tissue depots (e.g., the depot representing subcutaneous fat is not selected) does not necessarily mean that the two types of adiponectin have different metabolic functions. The infusion should be followed by at least a determination of a basic metabolic profile of the animals, e.g. at least blood glucose and insulin levels, but preferably GGT.

Since weight did not change significantly, while the weight of the fat depots was altered, this means that body composition must have changed, and it is not discussed. The change in weight must necessarily be accompanied by an analysis of food consumption and should be documented by weight curves for each group, which would also show the same initial weight of the animals in each group.  For Figure 6B and C, which describes the change in depot size, the correction of g to 100 g is given, which makes no sense.

Thus, overall, the experimental part of the in vivo experiments is very vague and its outputs are overstated, and either should be significantly supplemented or preferably not shown. Based on the data presented, it cannot be argued that the “data indicate that gACRP specifically increases lipid accumulation by adipose tissue while fQAcrp does not (line 64-65) or that  Our results indicate that gACRP promotes obesity (line 397)”

The reviewer may have surmised that this manuscript really consists of two parts. In the first (Figs 1-5) we establish that thrombin is the protease most likely responsible for adiponectin cleavage into globular adiponectin. We feel this is a comprehensive biochemical study and have used mutagenesis to confirm the cleavage site is sensitive to thrombin. While this finding is unique, it still begs the question whether it has biological meaning. The inability to clearly identify the individual actions of fAcrp and gAcrp over the 20 years since gAcrp was discovered is one that deserves attention, and the preparation of mAcrp provided us with the means to do this. Thus, we feel both parts of the story fit better together. Our focus on adipose tissue was based on the results we obtained, and we felt this would suffice to prove two points: one, that mAcrp is a useful reagent and could be employed in additional studies, and two, that fAcrp and gAcrp have distinct metabolic effects within the context of adipose tissue. However, the reviewer has requested additional data, which is now included. Specifically, a panel has been added to Figure 6 showing the weights of the inguinal fat pads in response to the treatments. This fat pad represents subcutaneous fat and shows that it is also increased by gAcrp. Likewise, a panel has been added for total fat, which is the sum of the 3 fat pads weighed at the termination of the study. New data for fasting glucose is provided in Figure 7. In contrast to the effects of gAcrp on fat pad size, it has no effect on glucose levels. Instead, fAcrp and mAcrp treatments were found to elevate fasting glucose levels. This additional information clearly establishes that fAcrp and gAcrp have distinct biological effects, with gAcrp linked to adipose tissue, while fAcrp is linked to glucose metabolism. With addition of the glucose data (Figure 7 & lines 367-370) we have made alterations in the Abstract (lines 25 & 27), Graphical Abstract and Discussion (lines 431-434; 507-510; 544-545) to highlight this information. This also led to a modification in the title of the manuscript. We feel these additions requested by the reviewer have clarified this aspect of the manuscript and support our proposed model.

With respect to other points made, we do not have the data on food composition beyond that the mice had free access to a pelleted commercial high-fat diet 60% kcal from fat; D12492, Research Diets, New Brunswick, NJ. Food consumption was not monitored and insulin was not measured. The change in weight (Figure 6 panel A) is based on the weight of each animal at the time of pump implantation, and therefore all values are relative to their initial weight at the start of the treatments. The fat pad weights shown in panels B-E (including the new panels for inguinal and total fat) present the absolute values and not the change over time. Since the animals are not equal in weight, the fat pad data are presented relative to body weight for the purpose of normalization. This is standard practice. Finally, the previous version of the manuscript did not include comments regarding muscle (lean mass) as we do not have any relevant data beyond total body mass in relation to this parameter. At the request of the reviewer we have brought up the issue of body composition changes caused by the treatments, but we do not feel there is enough information for more than some limited speculation. For this reason, the section on body composition added to the Discussion was kept short (lines 523-526).

  1. The authors expressed recombinant proteins in bacteria that are unable to provide the usual posttranslational modifications of adiponectin that affect assembly of the usual forms of adiponectin (Banga, A., et al. (2009). "Adiponectin translation is increased by the PPARgamma agonists pioglitazone and omega-3 fatty acids." Am J Physiol Endocrinol Metab 296(3): E480-489.) Thus, the results should still be verified on proteins expressed in eukaryotic cells

We agree with the reviewer that using recombinant proteins generated via baculovirus or a eukaryotic cell type would be better than bacterial proteins. We hope to do so in the next study we undertake.

  1. Why was rat plasma used when adiponectin processing was tested in vivo in mouse and in vitro in porcine cells? Is it known what thrombin activity the plasma used had (was it a pooled sample, or the plasma was obtained from different animals, in different pathophysiological conditions, e.g. endothelial damage, where a higher thrombin activity is expected)?

It has been clarified in the Methods (line 75) under fAcrp Cleavage Assay that the rat plasma, obtained from a colleague, was a pooled sample. The plasma worked in our system and we did not attempt to obtain any from another source.

  1. The text should clearly indicate whether the recombinant adiponectin was human. This information is now hidden in the references to previous work of the group.

This has been done in the appropriate section of Materials and Methods.

  1. If the authors performed a comparison of proteolytic activity between plasma, elastase, and trypsin, this result should be demonstrated on a single gel so that it can be compared side by side (Figure 1 A-D). Fig1 C,D-the authors claim that adiponectin was detected by western blotting, but the band pattern is more reminiscent of total protein detection by Gel Code Blue staining. Fig 1 A,B should include a line with plasma alone, where endogenous adiponectin is also present (albeit at a significantly lower concentration)

Unfortunately, at this point we are not able to complete the request of the reviewer due to the short time requested by the journal to complete this revision. We feel that the similarity of the banding pattern for panels A-D is sufficient to compare the results obtained with the different proteases.

  1. The identification of thrombin as the major protease of 30 kDa adiponectin was preceded by gel filtration of plasma, which indicated that the protease of interest was greater than 80 kDa. Therefore, trypsin and leukocyte elastase, as well as other possible proteases, were excluded because of their smaller size. However, the size of thrombin also reaches only 37 kDA, as the authors themselves state when presenting the results in Fig. 4D.  Somewhat purposely, they then claim that thrombin is usually in plasma in complex with other proteins- but this may be true for other proteases as well. Moreover, thrombin is most often in complex with antithrombin, which suppresses its proteolytic activity. Although I am not an expert on the coagulation pathway and properties of thrombin, I believe this fact should be mentioned, or it should be acknowledged that the results do not exclude other putative proteases to be able to cleave adiponectin.

We have clarified in the Discussion (lines446-447) that the molecular mass of native thrombin is at least 80 kDa (subunit size is 37 kDa) and can exceed 200 kDa when in a complex. This is nicely shown in the paper we have referenced [27] based on the native gel electrophoresis they present. Trypsin should not be present in plasma except in persons with pancreatitis. Leukocyte elastase forms a 1:1 complex with alpha1-anti-proteinase inhibitor, which exhibits a mass of 78 kDa [Baugh & Travis (1976) Biochemistry 15:836]. We feel that our gel filtration data in conjunction with the sequence of the cleavage site, which is the primary indicator of specificity (see lines 248-258), leave no doubt that thrombin is the protease responsible for generating gAcrp.

  1. Figs. 2 and 4 -representative images of the gels do not include some conditions, which are then shown in the graphs showing band intensities. This should be unified. Fig. 2E should be reduced in size to put the entire panel of Fig. 2 on one page.

Figure 2 has been modified so that all panels are on one page as requested. A graph of the band intensities resulting from treatment with Mg2+ and Ca2+ has now been added to Figure 2D. We feel the results of thrombin inhibition with hirudin and argatroban are evident visually without the need to present the data in a graph.